# Oblique Vibratory Surface Grinding—Experimental Study

**DOI:** 10.3390/ma16175819

**Published:** 2023-08-25

**Authors:** Grzegorz Bechcinski, Norbert Kepczak, Witold Pawlowski, Wojciech Stachurski, Paulina Byczkowska

**Affiliations:** 1Institute of Machine Tools and Production Engineering, Faculty of Mechanical Engineering, Lodz University of Technology, Stefanowskiego 1/15, 90-537 Lodz, Poland; grzegorz.bechcinski@p.lodz.pl (G.B.); norbert.kepczak@p.lodz.pl (N.K.); wojciech.stachurski@p.lodz.pl (W.S.); 2Institute of Materials Science and Engineering, Faculty of Mechanical Engineering, Lodz University of Technology, Stefanowskiego 1/15, 90-537 Lodz, Poland; paulina.byczkowska@p.lodz.pl

**Keywords:** vibratory grinding, surface grinder, Fourier analysis, surface roughness, surface waviness

## Abstract

The article reports the results of experimental study of vibratory surface grinding in the range of low excitation frequencies and variable directions of excited vibrations in the plane of the table, and investigates the effect of these directions on the roughness and waviness of the ground surface. The tests were conducted on a production surface grinder with a vibrating table on which the samples were mounted. The table made it possible to change the direction for the introduction of vibrations to the workpiece (longitudinally, transversely, and obliquely to the longitudinal feed of the table) and the parameters of the introduced vibrations, frequency and amplitude. In the course of the study, selected parameters of surface roughness and waviness of samples ground conventionally and with vibrations introduced on the workpiece were compared. The results show an improvement in the roughness and waviness parameters of the vibration-ground surfaces compared to surfaces ground without vibration (conventionally). The profile of the ground surface was subjected to Fourier analysis and the harmonic components of the surface shape of the ground samples were determined to characterize the effect of the introduced vibrations on the surface roughness. It was determined that the direction of vibration introduction, which is most favorable in terms of the parameters of the geometric structure of the ground surface, is the direction perpendicular to the longitudinal feed of the grinding table. In other directions of vibration introduction, the simultaneous effect of improving both parameters of the geometric structure of the ground surface profile was not obtained.

## 1. Introduction

Grinding is usually the last technological operation applied to a workpiece, and therefore deviations from the accepted assumptions concerning the geometry of the workpiece surface shape and the properties of the surface layer after this operation should be as small as possible. Therefore, it is necessary to design and carry out this process as accurately as possible to maintain all required dimensions and tolerances [1].

Vibrations during grinding that occur between the grinding wheel and workpiece pose a challenge in maintaining process requirements. Extensive research has been conducted to detect and eliminate this dynamic phenomenon, which affects the grinding wheel/workpiece interaction, leading to uneven wear of the grinding wheel and causing poor surface quality of the workpiece. However, deliberate introduction of vibrations into the grinding process and control of these vibrations can result in improved surface geometric structure (SGS) parameters.

Vibration-assisted machining processes are widely used, especially when machining hard-to-machine materials, resulting in lower machining forces, providing the possibility of using more efficient machining process parameters while maintaining high quality of the machined surface [2,3,4,5,6,7,8]. Theoretical analysis of the coexistence phenomenon for the vibration and machining process has been carried out for both machining and grinding processes [9,10]. However, the applications of this technological method described in these publications do not take into account low-frequency vibration-assisted grinding or the direction of vibration introduction, nor do they offer an analysis of their effect on the geometric structure parameters of the machined surface.

In the scientific literature, it is very difficult to find attempts at vibratory grinding using low excitation frequencies. The work of Ibrahim et al. presents the results of their research during vibratory grinding of Inconel 718 [11]. In the tests conducted, the value of the inductor frequency was set to 100 Hz, which resulted in a vibration amplitude of 130 µm. For vibratory grinding compared to conventional grinding, results showed that the average grinding force decreased by about 30–40%, the surface roughness Ra increased by about 20–30%, and the grinding wheel-wear decreased by about 25–35%.

The paper by Batako and Tsiakoumis presents the results of vibratory grinding tests on nickel alloys and hardened steel under dry grinding conditions and using the MQL method [12]. In the study, the vibration frequency of the inductor was set at 275 Hz, which resulted in a vibration amplitude of 15 µm. The variable parameters were feed rate, grinding depth, and grinding speed. The results showed that by introducing vibration into the grinding process, a 15% decrease in grinding forces and a 10% increase in surface roughness quality Ra may be obtained.

The study by Ewad et al. implemented a vibratory grinding control system during grinding of steel in the soft and hardened states [1]. In the study, the vibration frequency of the inductor was set at 100 Hz, which resulted in a vibration amplitude of 120 µm. The results show that the introduction of vibration into the grinding process brought positive effects by reducing the value of grinding forces from 30 to 45%, together with a reduction in the roughness of the vibrationally ground surface by up to 40%.

However, most scientific work is based on introducing supersonic (ultrasonic) vibrations into the grinding process [13,14,15]. The work of Sreethul et al. showed how to design and implement a fixture for introducing ultrasonic vibrations in the grinding process [16]. In the study conducted, the frequency of the inductor was set at 19,900 Hz, which resulted in a vibration amplitude of 12 µm. The results showed that for both grinding forces and surface roughness, an overall improvement in the grinding process was obtained.

The work of Singh and Sharma shows the introduction of supersonic vibration into the grinding process when machining a nickel–chromium alloy enriched with the addition of titanium, aluminum and carbon, known as Nimonic 80A [17]. In the experiments carried out, the frequency of the inducer was set at 20,000 Hz, which resulted in a vibration amplitude of 10 µm. The results show that the introduction of ultrasonic vibration had the effect of reducing grinding forces by up to 66% and reduced the roughness of the ground surface by 46%.

The work of Chen et al. shows the results of ultrasonic grinding of sapphire [18]. In the study, the frequency of the inductor vibration was set at 28,000 Hz, which resulted in a vibration amplitude of 6 µm. It was shown that with vibratory grinding, the grinding forces decreased by 10%, as did the grinding energy, while the surface roughness decreased by 12%.

The work of Zhang et al. shows the process of vibratory grinding of carbides [19]. In the study, the vibration inductor frequency was set at 35,000 Hz, which resulted in a vibration amplitude of 4 µm. The results clearly showed that the values of grinding forces (both tangential and normal) decreased by more than 21%.

The work of Li et al. presents a study of the ultrasonic-vibration-assisted grinding of difficult-to-machine materials (SiC ceramics) [20]. The varied-depth nanoscratch test of a single grain was carried out together with analysis of the grain trajectory of ultrasonic-vibration-assisted grinding. It resulted in acquiring a theoretical model of the normal grinding force. The ultrasonic-vibration-assisted grinding experiment for SiC ceramics was carried out in order to analyze the influence of the grinding parameters on the grinding force and to compare the differences between ultrasonic-vibration-assisted grinding and conventional grinding in surface and subsurface quality. The research results confirmed that appropriate vibration frequency and amplitude could improve the ground surface quality.

In most of the scientific work, the vibrations introduced into the system were in the direction parallel to the direction of the grinding-table feed. In contrast, the roughness parameter Ra was mainly studied. Nevertheless, in many countries and manufacturing plants, the preferred parameter that determines the quality of the ground surface is Rz, which gives the distance between the highest elevations and the deepest depressions of the profile [21]. Therefore, in this work, we focused on the introduction of variable directions of excited vibrations in the plane of the table into the surface grinding process, and studied the effect of these directions on the workpiece surface roughness and its waviness.

## 2. Materials and Methods

The test object is a SPC-20-type surface grinding machine equipped with a grinding wheel spindle bearing a sliding-track and a workpiece table driven hydraulically on sliding guides. The test stand for vibratory surface grinding is shown in Figure 1. To adapt the grinder to the requirements of implementing vibratory grinding, the machine tool was equipped with a device for introducing controlled vibrations to the workpiece. The design of the oscillating table is shown in Figure 2. The table is designed for clamping and for making the workpiece oscillate. On the base (1) (Figure 2) there are rolling rope guides (2) along with a chuck (3) holding the workpiece (4), which is moved. The oscillatory motion of the workpiece is implemented by means of an electrodynamic inducer (5). The electrodynamic inducer (5) makes it possible to obtain oscillatory vibrations of the table with a maximum peak-to peak amplitude of c.a. 600 µm and a frequency range of 20–10,000 Hz.

In the experimental studies, grinding tests without vibrations introduced into the process and with vibrations introduced into the ground workpiece were carried out for variable directions of excited vibrations in the table plane (0°, 45°, 90°) in relation to the table velocity vector (Figure 3). The effect of these directions of vibration introduction on the roughness and waviness of the ground plane surfaces was then analyzed.

The tests were conducted on specimens with a specially designed shape, a circular specimen with a diameter smaller than the width of the grinding wheel (Figure 4), made from C45 structural steel hardened to 54 HRC hardness.

In the vibratory grinding tests, a grinding wheel manufactured by Saint-Gobain Abrasives was used. It is a type 1-A-200x20x51-38A 60 G12 VBEP-33 grinding wheel and belongs to the very soft group. It has a large-core structure with electro-corundum abrasive grains and a ceramic bond.

To determine the amplitude of vibrations introduced into the surface grinding process using an oscillating table, it was necessary to develop its amplitude–frequency characteristics. Figure 5 shows the amplitude–frequency characteristics of an oscillating table set into vibration with an electrodynamic vibration inductor, modelPR 9270/01 from Philips. It made it possible to determine the maximum peak-to-peak amplitudes A_bl_, expressed in micrometers for the 20 to 500 Hz frequency range of the vibration inductor.

## 3. Results

During the experiments, the process of plunge peripheral rectilinear (tangential) surface grinding without and with vibrations introduced on the workpiece at angles of 0°, 45°, and 90° to the longitudinal feed direction of the grinding table was realized.

A typical grinding cycle commonly used in industrial practice consists of three phases:Pre-grinding (coarse): grinding depth 0.03 and 0.02 mm/pass;Finishing grinding (fine): grinding depth of 0.01 and 0.005 mm/pass;Spark-out grinding.

The dressing process was conducted as follows: 3 passes of 0.03 mm grinding depth, 3 passes of 0.02 mm grinding depth, 3 passes of 0.01 mm grinding depth, and 1 pass without in-feed introduction.

The following grinding process parameters and conditions were established:Longitudinal feed rate (2, 5, 10, and 15 m/min);Frequency of vibrations introduced (0, 20, 50, 100, 150, and 300 Hz);Spark-out grinding without and with introduced vibrations.

The following parameters of the ground surface were measured:Roughness (Ra and Rz parameters);Waviness of the ground surface (Wa_w_ parameter).

The profile of the ground surface of the specimen in the test samples was recorded with a Surftest-type instrument from Mitutoyo. The measuring section was *l_n_* = 4.8 mm.

### 3.1. Experimental Studies

Figure 6, Figure 7 and Figure 8 show the average values of the Ra, Rz and Wa_w_ parameters measured for the surface of the ground C45 steel specimen obtained when grinding without introduced vibrations (0 Hz) and when grinding with introduced vibrations of 20, 50, 100, 150, and 300 Hz. In addition, the direction of the introduced vibrations was changed during the experimental tests. Additional vibrations were introduced for angle 0°, in which the direction of the introduced vibrations was parallel to the longitudinal feed vector of the grinding table (Figure 6), for angle 45°, the direction of the introduced vibrations at an angle of 45° to the longitudinal feed vector of the grinding table (Figure 7) and for angle 90°, the direction of the introduced vibrations perpendicular to the longitudinal feed vector of the grinding table and at the same time parallel to the axis of the grinding wheel spindle (Figure 8). The longitudinal feed rate of the grinding table was also changed. The graphs marked (a) were obtained with the longitudinal feed of the grinding table *f =* 2 m/min, (b) *f =* 5 m/min, (c) *f =* 10 m/min, and (d) *f =* 15 m/min. Arrows showing the percentage change for a given surface condition parameter in relation to the initial value, i.e., obtained without input vibration, are also presented on the individual graphs. Arrows in red indicate deterioration of the surface condition, and arrows in green indicate improvement of the surface condition obtained after grinding with vibrations.

In the case of grinding flat surfaces with input vibration for angle 0° of the oscillating table setting on the turntable, in which the direction of the input vibration was parallel to the longitudinal feed vector of the grinding table (Figure 6), a reduction in the roughness parameters Ra and Rz was obtained over the entire range of input vibration frequencies, but for a small longitudinal feed of the grinding table *f =* 2 m/min. The maximum percentage improvement in the Ra parameter was 8.33% for a table vibration frequency of 300 Hz. For the Rz parameter, the maximum was 2.78% for table vibration frequencies of 100 and 150 Hz. At this low feed rate of the grinding table, a reduction in the waviness parameter Wa_w_ of 23.81% was obtained only for a table vibration frequency of 150 Hz. For the table feed rate increased to *f* = 5 m/min, only an improvement in the Ra and Rz parameter was obtained for table vibration frequencies of 20 and 50 Hz, with maximums of 10.7% and 18.6%, respectively. For a feed rate of *f =* 10 m/min, only a slight improvement in the waviness parameter Wa_w_ of 0.88% was obtained for a table vibration frequency of 100 Hz. On the other hand, for a feed rate of *f =* 15 m/min, an improvement in the waviness parameter Wa_w_ of 7.19% was obtained, also only for a table vibration frequency of 100 Hz.

In the case of grinding flat surfaces with vibrations introduced for an angle of 45° setting of the oscillating table on the turntable with respect to the longitudinal feed vector of the grinding table (Figure 7), a reduction in the roughness parameters Ra and Rz was also obtained over the entire range of vibration frequencies introduced for a small longitudinal feed of the grinding table *f =* 2 m/min. The maximum percentage improvement in the Ra parameter was 29.63% for a table vibration frequency of 20 Hz. For the Rz parameter, the maximum was 21.81%, also for a table vibration frequency of 20 Hz. At this low feed rate of the grinding table, a reduction in the waviness parameter Wa_w_ was obtained by a maximum of 20.63% for a table vibration frequency of 50 Hz and for table vibration frequencies of 100 and 300 Hz. For a table feed rate increased to *f =* 5 m/min, an improvement in the Ra and Rz parameters was obtained over the entire range of input vibration frequencies. The maximum percentage improvement in the Ra parameter was 37.21% for a table vibration frequency of 20 Hz, and for the Rz parameter, the maximum was 34.69%, also for the 20 Hz table vibration frequency. For a table feed rate of *f =* 5 m/min, an improvement in the waviness parameter Wa_w_ was obtained only for table vibration frequencies of 20 and 50 Hz, 8.33% and 28.13%, respectively. For a feed rate of *f =* 10 m/min, improvements in the Ra and Rz parameter were again obtained over the entire range of input vibration frequencies. The maximum percentage improvement in the Ra parameter was 28.4% for a table vibration frequency of 20 Hz. For the Rz parameter, the maximum was 17.52% for the 50 Hz table vibration frequency. An improvement in the waviness parameter Wa_w_ was also achieved for table vibration frequencies of 20, 50, and the 300 Hz (maximum 10.53%). On the other hand, for a feed rate of *f =* 15 m/min, an improvement in the Ra and Rz parameter was again obtained over the entire range of vibration frequencies introduced beyond the 150 Hz frequency (Ra decreased by 0.52%). The maximum percentage improvement in the Ra parameter was 12.5% for the 50 Hz table vibration frequency. For the Rz parameter, the maximum was 6.18%, also for the 50 Hz table vibration frequency. An improvement in the waviness parameter Wa_w_ was also achieved over the entire range of input vibrations, with a maximum of 39.22% for the table vibration frequency of 300 Hz.

In the case of grinding flat surfaces with vibrations introduced for an angle of 90° setting of the oscillating table on the turntable—the direction of the introduced vibrations perpendicular to the vector of the longitudinal feed of the grinding table and at the same time parallel to the axis of the grinding wheel spindle (Figure 8), a reduction in the roughness parameters Ra and Rz was also obtained in the range of vibration frequencies introduced from 20 to 150 Hz for a small longitudinal feed of the grinding table *f* = 2 m/min. The maximum percentage improvement in the Ra parameter was 27.78% for a table vibration frequency of 150 Hz. For the Rz parameter, the maximum was 19.58%, also for a table vibration frequency of 150 Hz. At this low feed rate of the grinding table, a reduction in the waviness parameter Wa_w_ was a maximum of 25.49% for the 150 Hz table vibration frequency and for the 100 Hz table vibration frequency. For the table feed rate increased to *f* = 5 m/min, an improvement in the Ra and Rz parameter was obtained over the entire range of input vibration frequencies beyond the 50 Hz input vibration frequency. The maximum percentage improvement in the Ra parameter was 26.36% for a table vibration frequency of 150 Hz. For the Rz parameter, a maximum of 27.43% was also achieved for a table vibration frequency of 150 Hz. For a table feed rate of *f* = 5 m/min, an improvement in the waviness parameter Wa_w_ was obtained only for table vibration frequencies of 20 and 150 Hz, 22.60% and 20.34%, respectively. For a feed rate of *f* = 10 m/min, an improvement in the Ra and Rz parameter was again obtained over the entire range of input vibration frequencies. The maximum percentage improvement in the Ra parameter was 44.44% for a table vibration frequency of 20 Hz. For the Rz parameter, the maximum was 36.86%, also for the 20 Hz table vibration frequency. Improvements in the waviness parameter Wa_w_ were also obtained for table vibration frequencies of 100, 150 (41.90%), and 300 Hz (maximum 42.10%). On the other hand, for a feed rate of *f* = 15 m/min, improvements were again obtained in the Ra and Rz parameter over the entire range of input vibration frequencies. The maximum percentage improvement in the Ra parameter was 30.23% for a table vibration frequency of 150 Hz. For the Rz parameter, the maximum was 22.85%, also for a table vibration frequency of 20 Hz. An improvement in the waviness parameter Wa_w_ was also achieved over the entire range of input vibrations, with a maximum of 49.67% for the 100 Hz table vibration frequency.

### 3.2. Fourier Analysis of the Surface Profile of Ground Samples

To carry out a detailed analysis of the effect of vibrations of the table introducing oscillations to the workpiece on the parameters of the geometric structure of the surface, the profiles obtained when grinding the sample without introduced oscillations and the profiles of the ground surface with introduced oscillations in the axial direction (perpendicular to the feed motion of the table during grinding of the surface) were considered. The basic grinding kinematic parameters and the grinding cycle were the same in both analyzed cases.

Fourier transformation is usually applied to waveforms carried out in the time domain, allowing the analysis of measurement results in the frequency domain [22].

For surface profile analysis, the primary domain of the measured waveform is length rather than time, and the measurement values define the shape of the measured surface along the measurement segment [23].

Fourier analysis involves decomposing the measured signal into a series of sinusoidal waveforms, the superposition of which gives the original waveform. The results of the analysis are values expressed as complex numbers. In the case where a surface profile has been considered, rather than a waveform in the time domain, the domain in which the results of Fourier analysis are described is the inverse of the period of the sinusoids into which the original signal was decomposed.

Figure 9 and Figure 10 present the results of Fourier analysis of the profile from a workpiece ground without oscillations (Figure 9) and the profile from a workpiece ground with axial oscillations (introduced perpendicular to the direction of table movement) with a frequency of 20 Hz and a peak-to-peak amplitude of c.a. 600 µm (Figure 10). Figure 11 summarizes the results of Fourier analysis for both of these cases in the range in which the introduced workpiece oscillations interacted, i.e., for wavelengths smaller than the peak-to-peak amplitude of the 600 µm oscillations (from 0.00167 to 0.2082 µm^−1^).

Comparison of the Fourier analysis results in waveforms shown in the figures above indicates that the amplitude of the spectrum (magnitude of Fourier transform) for the profile of the ground sample with axial oscillations for large wavelengths are at a much lower level than for the profile of the ground sample without oscillations. The reason for this observed phenomenon is the obliteration of the machining trace formed on the workpiece by the same grain put into axial oscillations. The grain, being in contact with the workpiece in the next rotation of the grinding wheel, does not exactly hit the trace created in its previous pass. The introduced axial oscillations move the position of the grain within the amplitude range of the workpiece oscillations and cause the trace generated by the abrasive grain to become wider and begin to interfere with the traces of neighboring grains. The intensity of the obliteration of the trace, and therefore the degree of mutual interference of the traces of adjacent abrasive grains, decreases with the inverse of the wavelength, that is, with the decreasing amplitude of the axial oscillations of the workpiece. This is directly related to the size of the abrasive grain used in the grinding wheel and the packing density of the grains on the working surface of the wheel.

In the case of the tests carried out, an abrasive wheel from Saint-Gobain with the designation 38A60G12VBE-33 was used. The abrasive grain size of this wheel is about 300 µm. This information was verified by structure images taken with a Keyence VHX-950F series digital microscope. The images were obtained by compiling individual images with different focus positions. Figure 12 shows one of the measurements taken from an image of the grinding wheel. The distances between adjacent abrasive grains were determined to be close to 300 µm.

The maximum theoretical density of abrasive grains along the roll formation of the grinding wheel when they are packed without a bonding layer cannot be greater than 300 µm. Correlation of the size of abrasive grains together with their theoretical packing density indicate the amplitude of axial oscillations, which specify that 600 µm is the most effective peak-to-peak amplitude of oscillations to reduce the roughness of the grinding profile. At this amplitude of oscillation, the trace of the abrasive grain on the surface of the workpiece between two adjacent abrasive grains is fully obliterated. This amplitude magnitude for the axial oscillations of the workpiece was therefore used in this study. The results of the experiments confirmed the suitability of the inference and gave results confirming the improvement in the profile quality of the ground surface by reducing the Rz coefficient of roughness.

The measured roughness of the profile, both Ra and Rz, on the measuring section takes into account the deviations in the height of the actual profile from the reference (flat) profile. In the results from the Fourier analysis of the surface profiles of the ground workpieces without and with axial oscillations, for a wavelength equal to the peak-to-peak amplitude of the axial oscillations of the workpiece, the most intense obliteration of the machining trace can be observed, which, as a result, explains obtaining lower Rz values in the conducted measurements for the surface of ground samples with axial oscillations compared to the surface of ground samples without oscillations.

## 4. Summary and Conclusions

The effect of vibrations introduced into the surface grinding process on the condition of the workpiece surface layer was determined by measuring the roughness parameters Ra and Rz and the waviness parameter Wa_w_ in the longitudinal direction.

From the analysis of the results of measurements of the roughness parameters Ra and Rz, it can be seen that an improvement in surface condition—a reduction in roughness—is obtained when the oscillating table is set at an angle of 45° to the direction of the grinder-table feed when grinding with vibration. However, the greatest improvement in surface roughness was obtained when the table is set at 90° and grinding with vibrations introduced at a frequency of 20 Hz.

In the case of measurements of the waviness parameter Wa_w_, its reduction was obtained at high feeds and input vibration frequencies of 20 and 50 Hz for an oscillating table setting angle of 45° and an input vibration frequency of 100 Hz, and for an oscillating table setting angle of 90° and an input vibration frequency of 150 Hz.

The reduction in the value of the surface waviness parameter may be due to the effect of variable stiffness of the workpiece clamping assembly. This results in the disruption of the process of regenerative self-induced vibrations on the surface of the workpiece.

The results of the experimental study do not indicate the need for an oblique setting of the table at an angle of 45°, which would have an effect on increasing the values of the roughness parameters more than when introducing vibrations at an angle of 0°, and on increasing the values of waviness parameter than when introducing vibrations at an angle of 90°. From the point of view concerning the parameters related to the geometric structure of the ground surface, the most favorable direction for introducing vibration is setting the vibrating table perpendicular to the direction of longitudinal feed for the adopted grinding conditions.

The reduction in the values of the roughness parameters of ground flat surfaces for samples with vibrations introduced at 45° and 90° angles is the result of kinematic superimposition of the oscillatory motion of individual abrasive grains on parallel machining lines. Thus, there is crossing of machining traces and leveling of unevenness in the sample surface profile. This is shown by the Fourier analysis performed on the surface profile of the ground samples.

## Figures and Tables

**Figure 1 materials-16-05819-f001:**
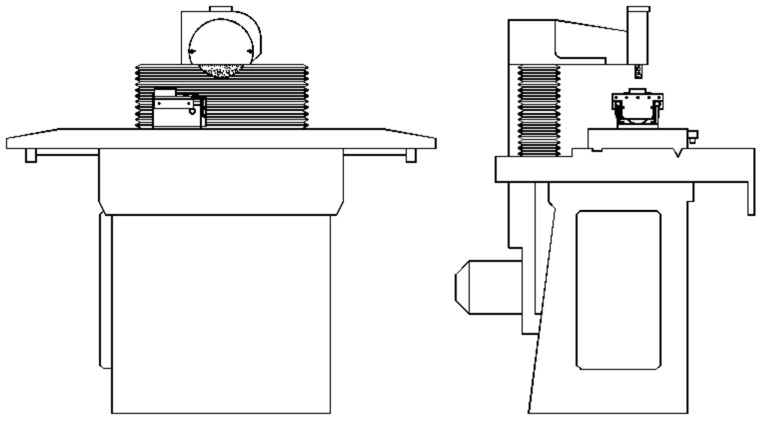
SPC 20 surface grinder.

**Figure 2 materials-16-05819-f002:**
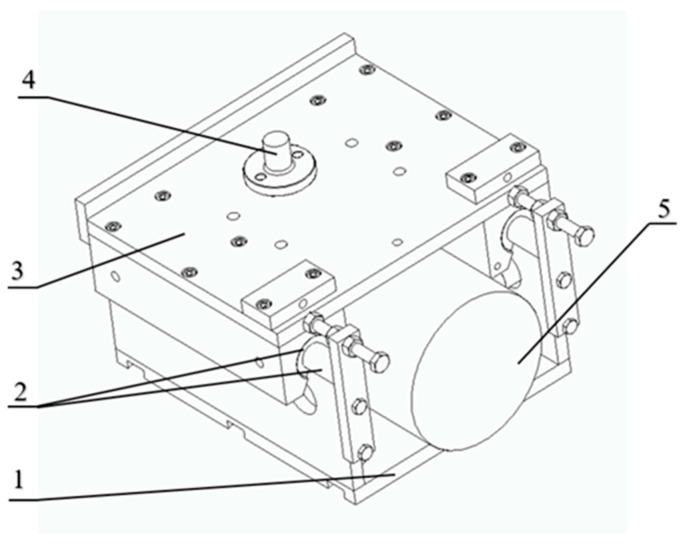
Oscillating table.

**Figure 3 materials-16-05819-f003:**
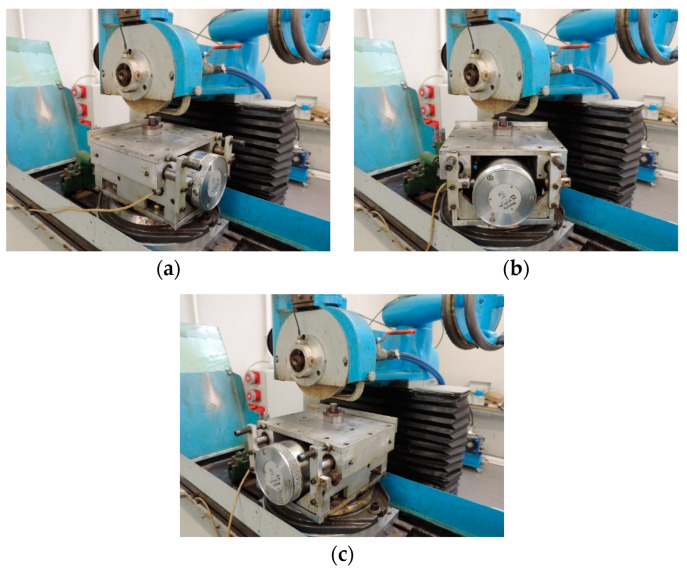
Orientation of the oscillating table on the grinding table: (**a**) angle 0°—direction of vibrations introduced parallel to the longitudinal feed vector of the grinding table; (**b**) angle 45°—direction of vibrations introduced at an angle of 45° to the longitudinal feed vector of the grinding table; (**c**) angle 90°—direction of vibrations introduced perpendicular to the longitudinal feed vector of the grinding table and at the same time parallel to the axis of the grinding wheel spindle.

**Figure 4 materials-16-05819-f004:**
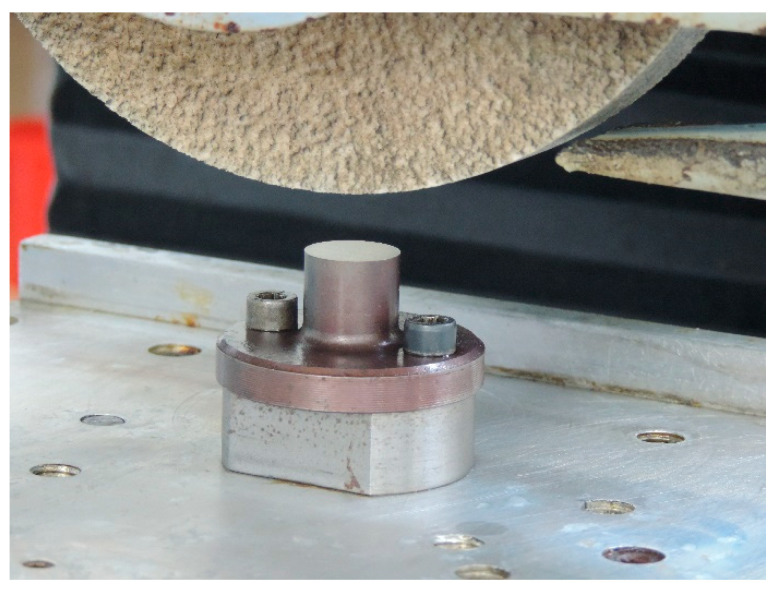
View of a ground C45 steel specimen mounted on an oscillating table.

**Figure 5 materials-16-05819-f005:**
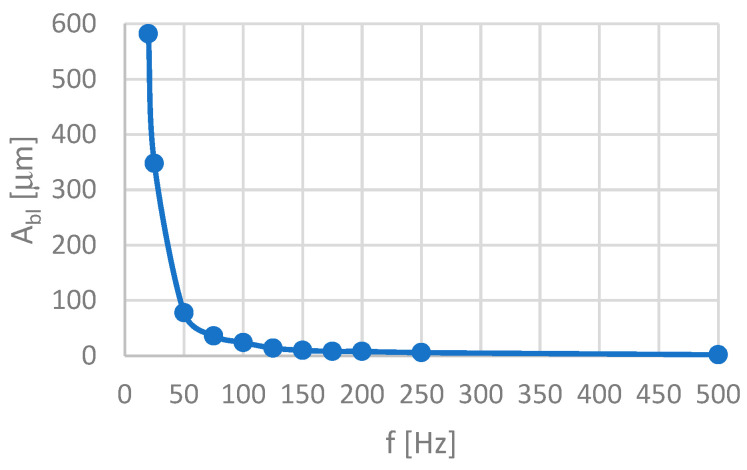
Amplitude–frequency characteristics of an oscillating table vibrated with an electrodynamic vibration inductor, model PR 9270/01 from Philips.

**Figure 6 materials-16-05819-f006:**
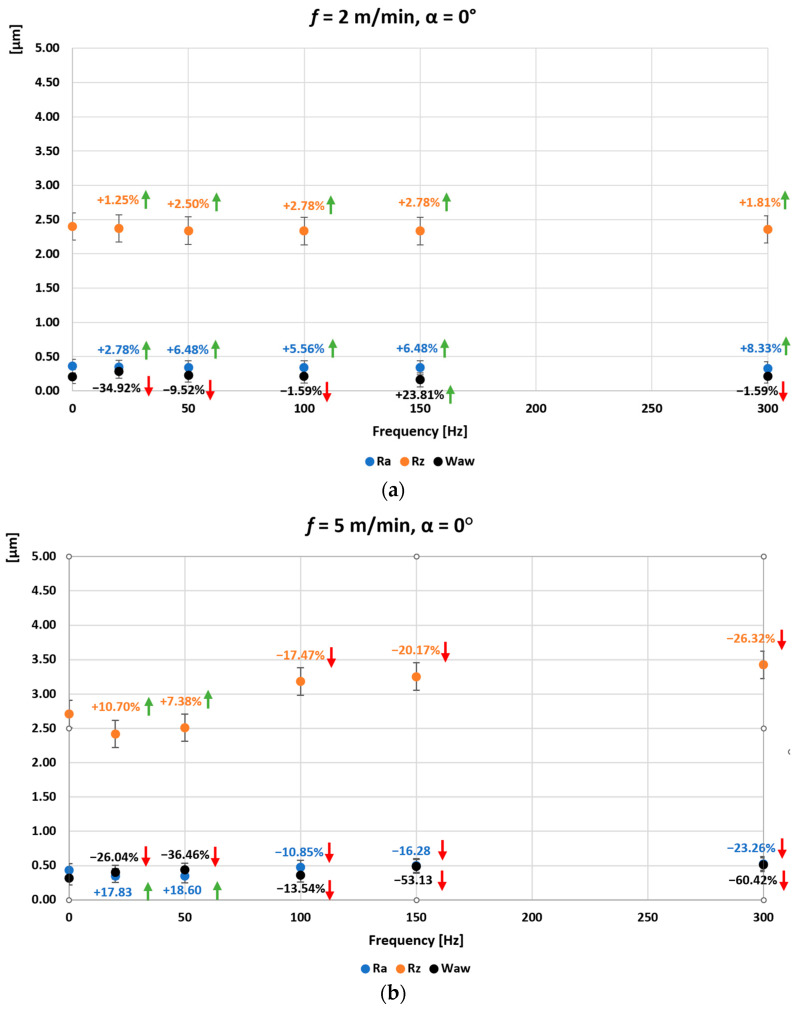
Average values of the parameters Ra, Rz, and Wa_w_ for the surface of the ground sample of C45 steel as a function of the frequency and direction of the introduced vibrations: angle 0° and the longitudinal feed of the grinding table (**a**) *f* = 2 m/min, (**b**) *f* = 5 m/min, (**c**) *f* = 10 m/min, (**d**) *f* = 15 m/min.

**Figure 7 materials-16-05819-f007:**
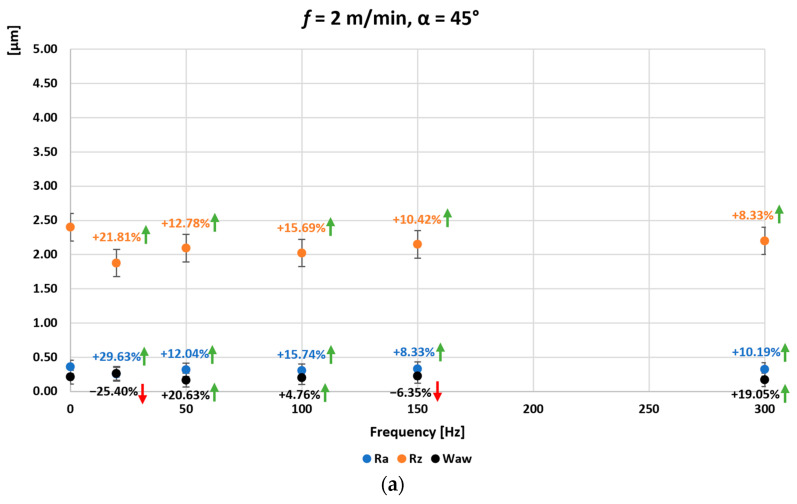
Average values of the parameters Ra, Rz, and Wa_w_ of the surface of the ground sample of C45 steel as a function of the frequency and direction of the introduced vibrations: angle 45° and the longitudinal feed of the grinding table (**a**) *f =* 2 m/min, (**b**) *f =* 5 m/min, (**c**) *f =* 10 m/min, (**d**) *f =* 15 m/min.

**Figure 8 materials-16-05819-f008:**
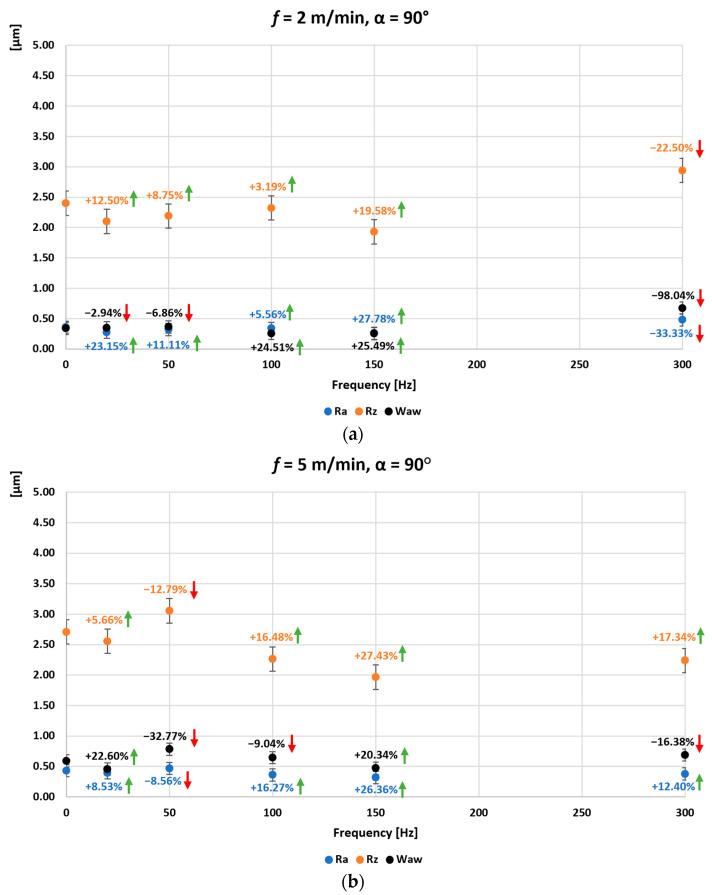
Average values of the parameters Ra, Rz, and Wa_w_ of the surface of the ground sample of C45 steel as a function of the frequency and direction of the introduced vibrations: angle 90° and the longitudinal feed of the grinding table (**a**) *f =* 2 m/min, (**b**) *f =* 5 m/min, (**c**) *f =* 10 m/min, (**d**) *f =* 15 m/min.

**Figure 9 materials-16-05819-f009:**
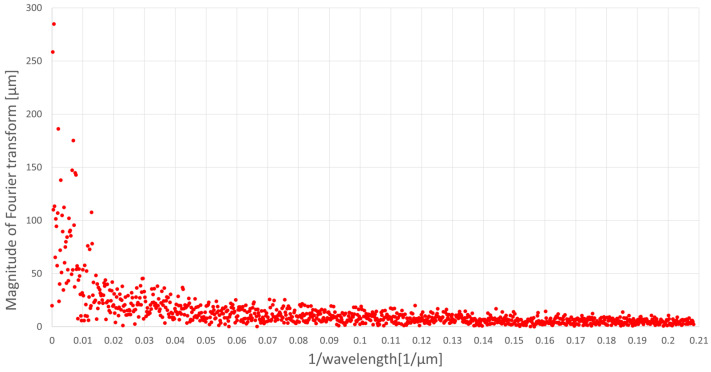
Fourier analysis of the profile ground without oscillations of the workpiece.

**Figure 10 materials-16-05819-f010:**
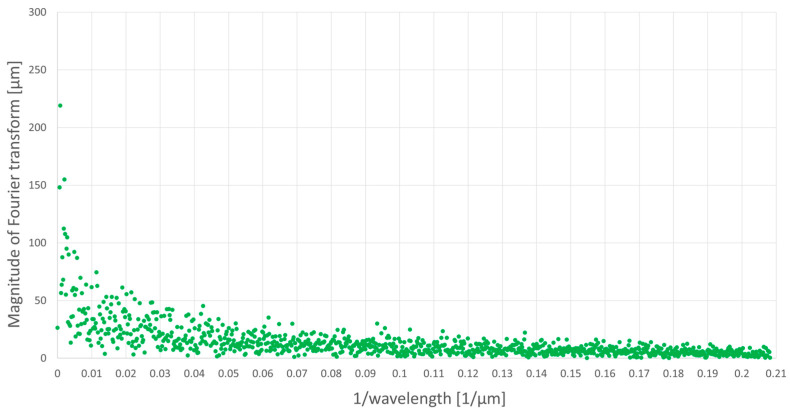
Fourier analysis of the profile ground with axial oscillations of the workpiece.

**Figure 11 materials-16-05819-f011:**
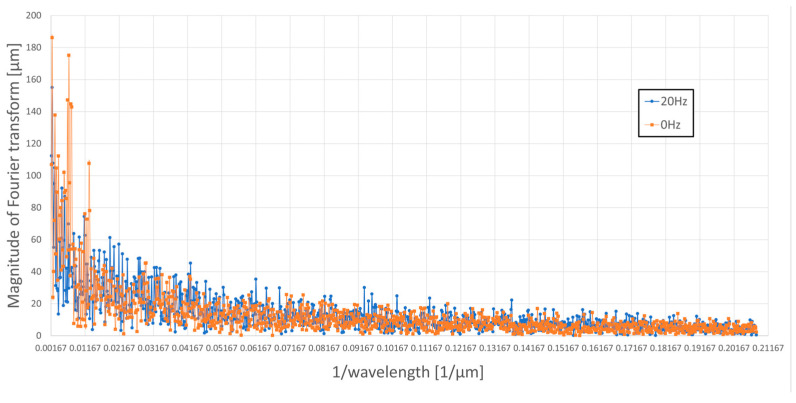
Fourier analysis of the profiles ground with and without axial oscillations of the workpiece.

**Figure 12 materials-16-05819-f012:**
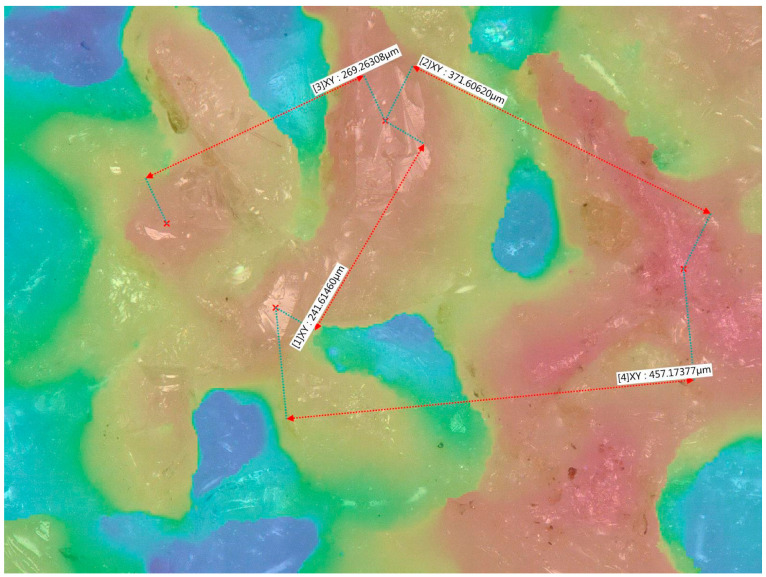
Microscope photography of the grinding wheel surface.

## Data Availability

The data presented in this study are available on request from the corresponding author.

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
