# Peer review of "Oblique Vibratory Surface Grinding—Experimental Study"

_materials, 2023, doi:10.3390/ma16175819_

Round 1

Reviewer 1 Report

This work performed vibratory surface grinding tests in the range of low excitation frequencies and variable directions of excited vibrations in the plane of the table, and the effect of vibration parameters on the roughness and waviness of the ground surface was investigated. The results indicated that the direction of vibration introduction is most favorable to improve the surface quality. Before publication, the authors should improve the manuscript by considering the following comments.

-All the experimental data should be added error bars.

-There is only description of experimental results of Figs. 6-8, and more explanation should be added in the manuscript.

-The optical images of the ground surface with and without vibration conditions should be added in the manuscript.

-The quality of Figs. 6-11 should be improved significantly.

-The following paper also studied the vibration grinding of difficult-to-machine materials, which indicated that appropriate vibration frequency and amplitude could improve the surface quality. The authors can refer to it in the manuscript.

Material removal mechanism and grinding force modelling of ultrasonic vibration assisted grinding for SiC ceramics. Ceramics International, 2017, 43: 2981-2993.

-

Reviewer 2 Report

Dear Author

the article is well written technically. However some suggesions for final approval are mentioned here

1. The results are discussion need to be more elaborate with proper referencing

2. The reason for improvement in surface finish for vibration assisted grinding is not clearly mentioned

Reviewer 3 Report

Interesting work that needs a little improvement:

1. Line 119 “frequency range of 20 Hz - 10 kHz” vs lines 143-144 „from 20 Hz to 500 Hz” and figure 5. What was the amplitude at f=0, 20, 50, 100, 150 and 300 Hz?

2. Figures 6…8 will be true using the column shape instead of linear shape, because was not used continuously variation of frequency between 0…300 Hz, only 6 steps.

Round 2

Reviewer 1 Report

The revised manuscript can be accepted.

Reviewer 2 Report

Dear Author

Corrections are satisfactory and can be processed further